# Tubes and bubbles topological confinement of YouTube recommendations

Camille Roth[1,2]*, Antoine Mazières[1], Telmo Menezes[1]

**1** CNRS, Centre Marc Bloch, Computational Social Science team, Berlin, Germany, **2** CAMS, Centre d'Analyse et de Mathématique Sociales, CNRS/EHESS, Paris, France

☉ These authors contributed equally to this work.
* roth@cmb.hu-berlin.de

**Data Availability Statement:** All data sets used in this paper are currently available at the following URL: https://www.nakala.fr/page/data/11280/fcd04889 Resource IDs have been mapped to

## Abstract

The role of recommendation algorithms in online user confinement is at the heart of a fast-growing literature. Recent empirical studies generally suggest that filter bubbles may principally be observed in the case of explicit recommendation (based on user-declared preferences) rather than implicit recommendation (based on user activity). We focus on YouTube which has become a major online content provider but where confinement has until now been little-studied in a systematic manner. We aim to contribute to the above literature by showing whether recommendation on YouTube exhibits phenomena typical of filter bubbles, tending to lower the diversity of consumed content. Starting from a diverse number of seed videos, we first describe the properties of the sets of suggested videos in order to design a sound exploration protocol able to capture latent recommendation graphs recursively induced by these suggestions. These graphs form the background of potential user navigations along non-personalized recommendations. From there, be it in topological, topical or temporal terms, we show that the landscape of what we call mean-field YouTube recommendations is often prone to confinement dynamics. Moreover, the most confined recommendation graphs i.e., potential bubbles, seem to be organized around sets of videos that garner the highest audience and thus plausibly viewing time.

## Introduction

The effect of algorithms in the filtering of information and interactions in online platforms is currently at heart of a very active debate, especially regarding the serendipity of contact and content discovery. On the one hand, a growing literature aims at empirically comparing what happens when users do rely, at least in part, on the output of some recommendation algorithm *vs.* when they do not. This kind of scientific endeavor generally need not venture into knowing or reverse-engineering which principles drive these algorithms. Contrarily, perhaps, to intuitions related to the popularization of so-called "filter bubbles", several recent studies appear to show that algorithmic suggestions do not necessarily contribute to restrict the horizon of users. Be it in terms of interaction or information consumption, users do not seem to be proposed less diverse content in regard to what would happen in the absence of recommendation

anonymized integer values, consistent across all datasets.

**Funding:** This work was partially supported by the "Algodiv" grant (ANR-15-CE38-0001) from the ANR (French National Agency of Research https://anr.fr) and the "Socsemics" Consolidator grant from the European Research Council (ERC https://erc.europa.eu) under the European Union's Horizon 2020 research and innovation program (grant agreement No. 772743) both awarded to CR. The funders had no role in study design, data collection and analysis, decision to publish, or preparation of the manuscript.

**Competing interests:** The authors have declared that no competing interests exist.

[1–6] or by using distinct recommendation approaches [7, 8]. The notable exception stems from explicit personalization i.e., explicitly chosen [9], or self-selected [10], by users [11]. Put shortly, the picture that seems to emerge is that filter bubbles, and possibly echo chambers, mostly occur when platforms recommend content based on explicit personal preferences (e.g., by subscribing to channels, specifying lists of interests, etc.) rather than on implicit usage traces or histories (either at the user-level or aggregated from the activities of all users).

On the other hand, at a more downstream level, user reactions to algorithmic curation are an equally important issue. The current state of the art exhibits mixed results. For one, user populations may not be deemed to be homogeneous: users may variously seek diversity [12], be variously responsive to recommendation [1], use it for various purposes [13] or have various expectations about it [14]—in these respects, the "average user" does not really exist. Users however seem to be generally sensitive to social signals and goaded by the indication that some content is popular or appreciated [15–17], whereas they are weakly sensitive to content-based signals, for instance if they are informed of the diversity of what they are currently consuming [18]. Such studies generally require the design of sophisticated experimental protocols or privileged access to private company data. On the whole, there appears to be no blanket answer to the complex interplay between the structure of proposed recommendations and user attitudes towards them.

In this context, while YouTube has become a key content provider (being the second most popular site as of 2019), the influence of its recommendation system on user navigation dynamics and exploration diversity has been little studied (even though it is already a current news topic, see e.g., [19, 20]). The present contribution intends to bridge this gap by focusing on the global, platform-level and thus non-personalized recommendations of YouTube. Indeed, irrespective of the personalized, user-centric adjustments to recommendation, studying platform-level suggestions shall provide an overview of the forces that are susceptible to apply globally to all users. As such, characterizing a possible confinement on these recommendation landscape constitutes a primary step toward characterizing confinement as a whole.

On content-sharing platforms, a model of user behavior toward recommendations may be construed as the navigation on a recommendation graph where nodes are items (such as videos) and links are recommendation suggestions, which users may or may not follow. Understanding the heterogeneity of the subsequent navigation topologies is crucial to appraise possible confinement processes. The issue of potential navigation topology has already generated several key studies in other platforms such as Twitter or Facebook, especially with respect to polarization and fragmentation, yielding convincing graph typologies (see, inter alia, [21–25]). By contrast, the state of the art relevant to YouTube's algorithms appears to have essentially focused on their technical underpinnings [26], their improvement [27] or their impact on consumption and audience statistics [28–30]. To our knowledge, very few academic works appear to focus on the structure of the browsing network: [31] describes the potential navigation dynamics in relation to audience or macro-level features, [32] principally uses the recommendation graph as a data source for extracting crowdsourced content taxonomies, while [33] focuses on the reachability of a specific portion of radicalized content by following commenting users, related channels or videos.

The paper is broadly organized as follows. We first carry out an instrumental step by exploring how video recommendation sets are being provided by the platform in the absence of personalization. This enables us to devise a robust protocol of collection of node-centric recommendation graphs. We then use these graphs to determine whether recommendation on YouTube tends to lower or increase the diversity of consumed content, and under which conditions. We thus analyze the shape of the graphs that are thus generated and their various confinement features. Most importantly, we discuss them in relation to various intrinsic

properties of videos (especially in terms of popularity, consensus, or topics), both in a static and longitudinal manner. This enables us to describe YouTube as an exception in the emerging state of the art on the effect of algorithmic recommendation on diversity, which seems to convene that platforms expand rather than limit the navigation horizon of users.

## Node-centric analysis of recommendations

Most YouTube videos include a tab featuring a list of suggested videos. How user-specific these suggestions are depends on whether users are logged in or share cookies and other identification information. While some suggestions seem to be clearly *user-centric* and depend on user navigation history (generally labeled by YouTube as "*Recommended for you*"), others appear to be *node-centric*, i.e. stemming from a pool of suggestions attached to the video itself, independently of the user history. In this latter case, suggested videos most likely depend on inferences made from platform-level behavioral traces accumulated over an unknown pool of users and an unknown period of time. This engenders a dichotomy between user-specific suggestions and what may be called a "mean field" of user-independent suggestions. We aim at characterizing this mean field, while leaving personalized recommendation outside of the scope of this paper. We do not aim at all at reverse-engineering the way node-centric suggestions are being computed by the platform, but rather wish to understand the navigation landscape that YouTube algorithms contribute to shape. In other words, we take for granted how these recommendations are built and focus on characterizing this landscape. Users are admittedly exposed to both types of suggestions, yet we contend that the analysis of the mean recommendation field is already likely to shed light on the attraction forces exerted by node-centric suggestions, all other things being equal.

### Data

In practice, we thus study user-independent suggestion lists attached to videos by creating non-persistent, anonymous sessions with simple HTTPS requests on a given page from a set of about a hundred IP addresses located in the region of Paris, France. This publicly available data collection step has been carried out in accordance with YouTube Terms of Services, especially its "Permissions and Restrictions" section, paragraph 3(c), as per EU Directive 2019/790 and its Article 3(1) on data mining for the purpose of scientific research (anonymized datasets and corresponding scripts for this paper have been made available in [34]). We first define a diverse set of YouTube videos by arbitrarily selecting five distinct sets of sources which feature links to such videos. Two of these sets aim to capture mainstream use by focusing on "Top" videos listed on Reddit and Wikipedia. The first set, denoted as "Reddit top" consists of the YouTube URLs contained in the most voted-up posts of 20 of the most subscribed subreddits (i.e., forums) listed on *redditlist.com*. The second set gathers the 5 most viewed videos from the 50 YouTube channels listed on the "List of most-subscribed YouTube channels" Wikipedia page [35], which we denote as "Wikipedia Top". This yields a diverse range of popular content mainly categorized by YouTube as "Music", "Entertainment", "Howto & Style" or "Science & Technology". The remaining sets focus on the activity surrounding the 2019 European Parliament election. While not as popular or representative of the use of YouTube, focusing on this context also contributes to reach political and election-related content. More precisely, the third set, denoted as "Twitter", consists of the most shared YouTube video links found on the micro-blogging platform over the 3 weeks leading to the 2019 European Parliament Elections and associated with a set of 22 hashtags that were manually selected to cover discussions related to the elections in 4 European languages: English, French, German and Italian (such as #EuropeanElections2019, #Europeennes2019, #Europawahl2019, #elezionieuropee, among

**Table 1. Seed categories and basic statistics.**

| Seed set | Seeds | *min* | Views | *max* | Top categories ≥10% (using YouTube labels) |
|---|---|---|---|---|---|
| | | | *median* | | |
| Reddit Top | 178 | 2,102 | 1,595k | 228m | Entertainment (15.7%), Science & Technology (12.9%), People & Blogs (12.9%), Howto & Style (11.8%), Music (10.1%) |
| Wikipedia Top | 161 | 528,159 | 91,430k | 4,242m | Music (34.8%), Entertainment (22.4%) |
| Political DE | 73 | 299 | 111k | 1.25m | News & Politics (100%) |
| Political FR | 88 | 116 | 33k | 0.8m | News & Politics (86.4%) |
| Twitter Top | 184 | 20 | 10k | 34m | News & Politics (68.5%) |

others). The fourth and fifth sets are based on channels of political parties engaged in the 2019 European elections in France and in Germany. In Germany, we focus on the 13 political parties that obtained a seat after the vote. In France, we equivalently consider the 13 main parties in descending order of obtained votes. In both cases, we identify the ten most viewed videos on each channel. We denote these seed sets respectively as "Political DE" and "Political FR". This political selection yields a more homogeneous set of videos than the mainstream ones: the three subsets consist of content that is principally categorized by YouTube as "News & Politics".

Table 1 gathers basic statistics for these seed sets, which comprise a total of 684 videos. Server errors, videos deleted during the data acquisition process and a handful unidentified crawling and parsing errors explain the discrepancy between the number of videos targeted for each set and the number of actually extracted seeds. While selecting IDs in a purely random manner across the platform could yield a more uniform sampling of video IDs on YouTube, it could bear the potential risk of overemphasizing insignificant videos with an extremely limited audience (using a protocol similar to [29], we verify that this would indeed be the case: a selection of 50 such random videos yields a very low median number of views of 115, far below the seed categories considered here).

Each HTTP request for the recommendations attached to a YouTube video returns a set of a maximum of 20 suggestions (in practice, exactly 20 suggestions four fifth of the time, and 19 about a fifth of the time). Our first conception of a model of a user navigating through these node-centric recommendations would thus consist of a walk in a directed recommendation graph whose nodes all have an out-degree of 19-20. However, for a given video, this set appears to fluctuate significantly from a request to the other, bearing the risk of exploring a very unstable and thus unreliable recommendation graph: examining the temporal features of these suggestions is thus a prerequisite to construct such a graph.

To this end, we proceed with a *long crawl* centered on seeds and aimed at understanding the variation and potential evolution of suggestion sets across successive requests. Along the way, we also collect video metadata such as the number of views and appreciation statistics: number of thumbs up (likes), down (dislikes). For each seed, we carry out a total of 2,000 requests at a regular average interval of about 10 minutes, thus covering a bit less than two weeks of sampling. This yields a node-centric time series of sets of suggestions.

## Stability of a recommendation plateau

We first compute the frequency of occurrence and recurrence aggregated over a certain number of requests $R$ in order to appraise the stability of suggestions and thus of the related network. Irrespective of the sampling duration, yet even more so for shorter time spans, a "plateau" of consistently highly frequently suggested videos quickly emerges (several of them

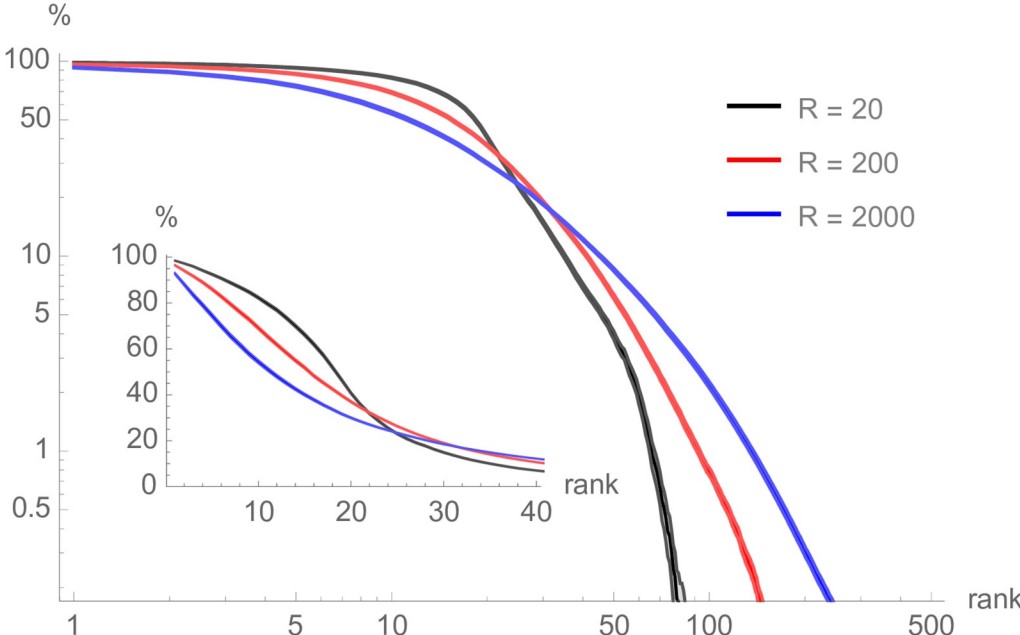

**Fig 1. Occurrence frequencies of suggested videos after *R* sampling requests, ordered by rank, averaged over all seeds.** *Inset:* zoom on the inflection area typically occurring around the 20[th] suggestion for *R* = 20.

are often recommended nearly 100% of the time), beyond which occurrence frequencies decrease steeply. The size of this plateau may be dynamically determined through a simple change-point analysis restricted to suggestions appearing at least, say, 1% of the time. This lower bound does not significantly change the position of the detected change point but is needed to prevent the very flat long tail of the distribution to interfere with the detection process. The plateau is generally found to feature between around 20 and 30 videos ($\mu$ = 23.6, $\sigma$ = 5.15). Nonetheless, its erosion over time suggests the existence of a slow renewal process. In the longer term, the ordered distribution of occurrence frequencies progressively takes the shape of an heterogeneous distribution apparently exhibiting a power-law-like tail with a cut-off. In Fig 1, we gather the frequency of occurrence of videos with respect to their rank, for various durations of aggregation. For instance, we see that the tenth most frequent suggestion after *R* = 200 requests (i.e. over about 33.3 hours, red curve) appears about 70% of the time. For all sampling durations *R* = 20, 200 and 2000, and all the more so for the shorter ones, occurrence frequency is relatively high up to the $\sim$20th most frequent suggestion and then markedly decreases afterwards. This suggests that exit routes leaving from a given seed and, thus, the recommendation graph induced by mean-field suggestions, are rather stable when observed on a relatively short time span of a couple of days.

To further qualify this observation, we turn to the study of the lifespan of suggestions. For a given seed video, we define the occurrence frequency of a suggested video *s* over a sliding window of *r* sampling requests as $\theta(s)$. We fix *r* = 20, consistently with the above-observed minimal amount of requests needed to observe a robust plateau. We then denote as *T(s)* the lifespan of *s*, defined as the difference between the first and the last moment where its average occurrence frequency $\theta(s)$ goes above a certain threshold. Put simply, the lifespan of a suggestion is such that it appeared above this threshold frequency at two moments separated by such a length of time. This does not mean, however, that it appeared consistently above this threshold over that length of time. We plot the numbers of suggestions having a lifespan of at least *T*

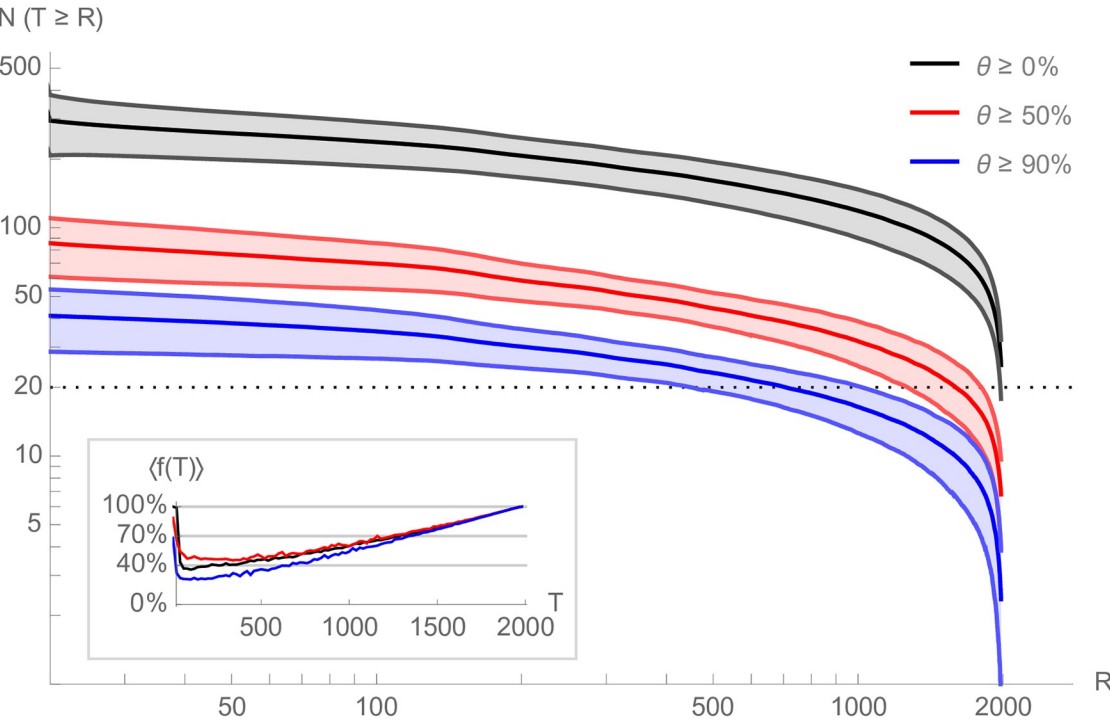

**Fig 2. Recommendation lifespans.** Number of recommendations with a lifespan $T \geq R$ for various thresholds $\theta \geq 0\%$, $\theta \geq 50\%$ and $\theta \geq 90\%$} (averages are central lines, along with their 95%-confidence intervals). A lifespan of $T$ means that a recommendation appeared at least $\theta\%$ of the time over a sliding window of $r = 20$ successive requests, at two distinct moments at least $T$ requests apart. *Inset:* average presence of a suggestion over its lifespan as a function of the lifespan (again for the three thresholds). Suggestions with longer lifespans are generally appearing very frequently during their lifespan ($T \to 2000 \Rightarrow \langle f(T) \rangle \to 100\%$).

(instead of exactly $T$, since we ignore what happens before or after we started collecting data). Fig 2 shows the distribution of lifespans for various thresholds: $\theta \geq 0\%$ corresponds to suggestions appearing at least once (i.e. at all), whereas $\theta \geq 90\%$ focuses on very dominant suggestions which appear at least 90% of the time over their lifespan and thus principally belong to the plateau. While this graph exhibits a relatively large number of short-lived suggestions, it also demonstrates that the plateau videos are likely to be present for a significant time. This is all the more the case as suggestions with higher lifespans also appear more frequently across their lifespan and not just at its extremities, as demonstrated by the inset in Fig 2.

This bears two conclusions when considering the recommendation graph induced by suggestions. First, focusing on the plateau would suffice as it concentrates most of the density of suggestion occurrence. This plateau has a modal distribution size and thus entails a network with a modal, homogeneous degree distribution—a quite peculiar object with respect to classical web topologies, which are generally heterogeneous. Second, this graph should be relatively stable in the short term, which substantiates the idea that a graph exploration protocol spanning over a short period would plausibly approximate well the recommendation graph faced by users during a navigation session.

## Induced recommendation graphs

For each seed video, we now recursively crawl suggestions belonging to the above-evoked plateau computed by changepoint analysis for 20 requests. We repeat this until reaching a depth of 3, which constitutes the horizon we consider for recommendation graphs. In other words,

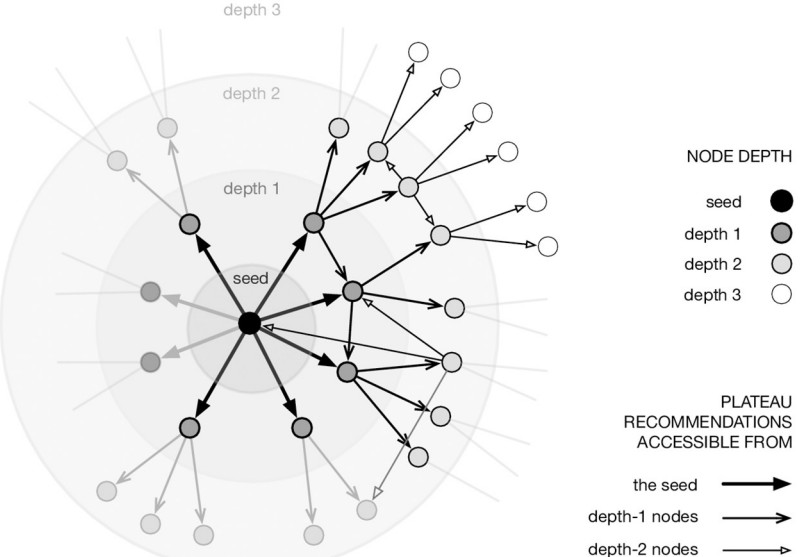

**Fig 3. Illustration of the recursive crawl focused on a given seed video.** Recommendations are crawled for the seed until a plateau may be estimated, which defines the direct neighbors of the seed and a set of nodes at depth 1. This process is repeated for all nodes at depth 1 in parallel, thus defining depth 2, and again with nodes at depth 2. In the end, the recommendation graph induced by the seed contains nodes at depths 1 and 2 and potentially includes links towards already explored nodes, i.e. at depth 0 (seed), 1 (seed's direct neighbors) and 2 (seed's indirect neighbors). There are on average 23.6 nodes at depth 1 ($\sigma = 5.15$), 325 nodes at depth 2 ($\sigma = 98.23$) and 2830 nodes at depth 3 ($\sigma = 1160$). Some elements are shaded simply to indicate that we do not represent all nodes and links on this figure for the sake of clarity.

the graph induced by a seed video contains its direct suggestions and two levels of indirect suggestions, as well as all the links between these nodes. See an illustration in Fig 3. Choosing an arbitrary depth of 3 is a trade-off between sampling frequency (to keep a reasonable bandwidth with YouTube servers) and sufficiently deep exploration of the various graphs. They are each collected in about 58.2hr (±13.6) which roughly remains within the plateau stability window (this corresponds to the time elapsed after about 350 requests in Fig 2). Graphs contain an average of 3179 ±1258 nodes and 7263 ±2233 edges. We crawled plateaus from nodes up to depth 2 (i.e., for around 89.0k videos) and thus visited nodes up to depth 3 (reaching a total of 540k videos).

## Graph entropy, diversity and confinement

We are specifically interested in exploring confinement within recommendation graphs. To this end, we devised two metrics. The first one is based on random walks, which play the role of a very simple and abstract model of user navigation (e.g., [25], to describe graph families, or [36] to quantify the diversity of user activity on online music platforms). Random walks always start from ego (the seed video of the induced recommendation graph), and terminate when they reach a length of 20. Results were not very sensitive to this constant, unless it is so small that meaningful walks can no longer be captured (<5). Other plausible random walk strategies include a restart once revisiting a node, or a restart once revisiting ego. Again, we found very similar results under such strategies, so we settled for the simplest one. We measure the diversity of visited nodes by computing the information entropy of the set of frequencies of visits. For one random walk, we refer to this measure as $\eta$. For each graph we

perform 100, 000 random walks—again, a value chosen to be high enough so that the results are stable across runs. The mean random walk entropy ($\bar{\eta}$) gives us an estimation of the confinement of an idealized user exploring the recommendation graph from ego. The lower the entropy / diversity, the higher the confinement. Another metric that we consider is the number of nodes in a recommendation graph ($N$). This configures a direct measure of the number of video recommendations that can be accessed from ego while not exceeding our maximal depth, independently of the probability of a user reaching a given node. Given that all out-degrees are roughly equal to 20 and maximal depth is 3 for all graphs, $N$ becomes indeed smaller when the set of targets accessible from the graph exhibits redundancy. To summarize, the first metric measures the propensity for diversity from the perspective of an idealized user following recommendations, and is determined by the topology of the graph. The second metric measures the global potential for diversity of the graph, independently of user behavior, and is simply determined by the size of the set of recommendations induced by a seed up to a certain depth.

In Fig 4 we show that the two metrics are negatively correlated ($\rho = -0.71$). This is somewhat counter-intuitive: it means that *the more diverse the mean random walk is, the less diverse the graph is, overall.* The dots in the scatter plot are colored according to the number of views of ego on a log scale. The darker the dot, the more views ego has. This helps illustrate another interesting fact: number of views are positively correlated with mean random walk entropy ($\rho = 0.36$) and negatively correlated with the number of nodes in the graph ($\rho = -0.44$). All of these correlations have a p-value $<0.0001$. It appears that videos with more views correspond to more "compact" recommendation graphs i.e., with a significantly smaller number of nodes while the diversity of the mean random walk is higher. We first provide illustrative visualizations of three sample graphs, corresponding to the closest graphs to the two extremes and the middle point of the regression line. These sample graphs provide a preliminary intuition of how topology changes across the spectrum defined by the correlation line.

Higher random walk entropy thus corresponds to smaller graphs, as well as denser graphs: there is a strong correlation between $\bar{\eta}$ and $\langle k \rangle$, the average degree of the graph ($\rho = 0.82$). These smaller and denser graphs exhibit higher connectivity—in the sense that everywhere is more accessible from everywhere else: even if the number of potentially accessible videos gets smaller (as graph size $N$ decreases), the number of actually accessible videos is higher (as further exemplified by the very strong correlation between $N_V$ and $\bar{\eta}$). Put differently, graph compactness nevertheless results in more isotropy in a smaller space: graphs with higher entropy lead to more videos being visited on average (higher $N_V$) whereas they stem from a smaller potential selection (smaller $N$).

Furthermore, we could confirm that graphs with higher $\bar{\eta}$ (i.e., more diverse random walks while having a smaller $N$) do also qualitatively appear to users to be more confined semantically. To substantiate this empirically, we designed a simple human-based protocol. We produced three sets of 20 seed videos which are closest respectively to each extreme and the middle point, similarly to the above procedure. We recruited six participants: each of them received plateau recommendations for 20 seed videos randomly selected among the 60, without knowing anything about them. We then asked them to tell us, for each seed, whether plateau videos are similar to one another or not, on a scale of 5 stars, from most similar (*****) to least similar (*). We gathered the aggregation of their subjective evaluation of the semantic confinement of plateau videos in Fig 5. We see that region 1 videos were perceived as most confined, while region 3 videos were seen as least similar, thus confirming a link between $\bar{\eta}$ and semantic confinement.

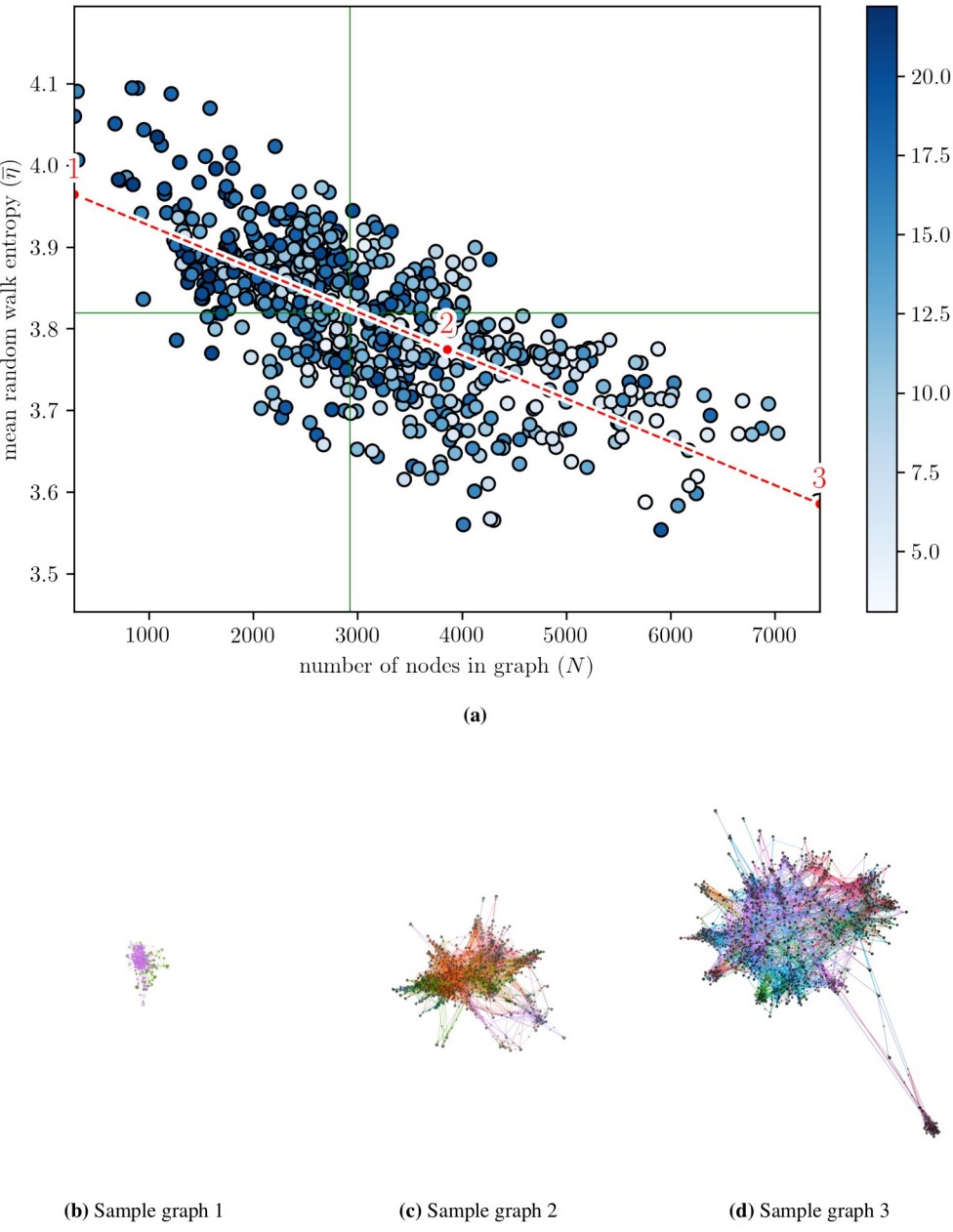

**(b)** Sample graph 1 **(c)** Sample graph 2 **(d)** Sample graph 3

**Fig 4. Induced recommendation graphs and sample visualizations.** (a) Induced recommendation graphs plotted according to number of nodes in the graph ($N$) and mean random walk entropy ($\bar{\eta}$). Points are colored according to number of views, on a log scale presented on the right. Solid green lines indicate medians, red dashed line is a linear regression of the distribution. Three points are marked in this latter line: one at each extremity and one in its middle. The three sample graphs (b), (c) and (d) are the closest ones to the three points indicated in the regression line of plot (a). Nodes (and adjacent edges) are colored according to the category of the video they correspond to.

## Confinement and seed properties

To expand our empirical exploration of confinement, we consider a number of other metrics. For the seed videos, we consider their age in seconds ($a$), their number of likes ($l$) and dislikes ($d$), and the number of subscribers ($s$) of the channel (i.e., video author) that they belong to.

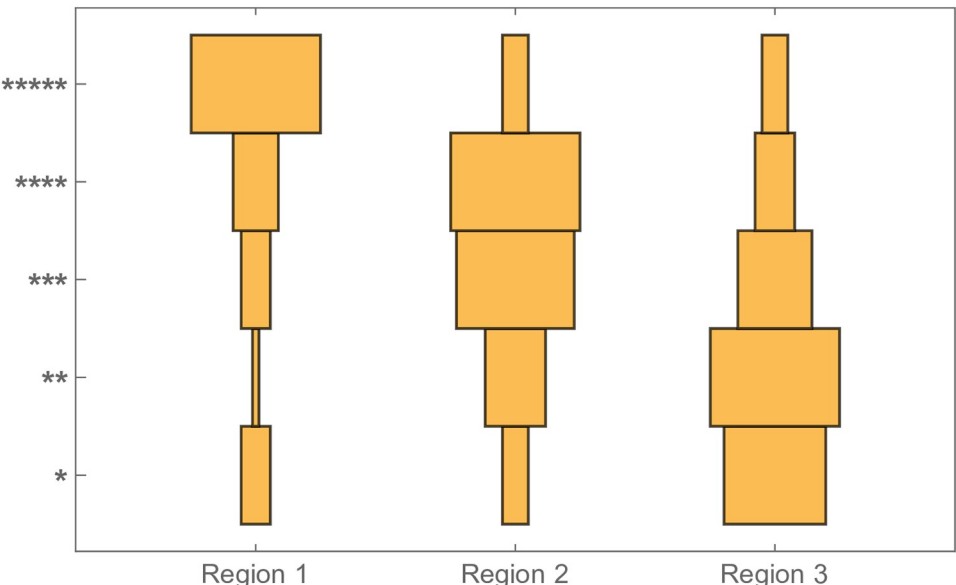

**Fig 5. Human evaluation of confinement.** Plateau recommendations for seed videos stemming from region 1 (largest entropy $\bar{\eta}$) are generally perceived as most similar (five stars), while the opposite holds for region 3. Region 2 appears as a middle way.

For the recommendation graphs, we apply the same random walk strategy to measure confinement or diversity in terms of video authors ($\bar{\eta}_a$) and categories ($\bar{\eta}_c$), as provided by YouTube.

In Fig 6 we present the correlations found between the above-mentioned metrics as well as the two original diversity metrics ($\bar{\eta}$ and $N$) and the number of views ($v$). It can be observed that all metrics that correspond to explicit user actions ($v$, $l$ and $d$) are highly inter-correlated, and also highly correlated with the number of subscribers ($s$) of the channel of the seed video, hinting at an audience effect. By "audience effect" we mean that the plausible existence of channel-dependent audiences (including subscribers) influences the viewership of a given video. To evaluate consensus around a video, we also derive from $l$ and $d$ a contentment index ($c$), computed as the log of the ratio of the number of likes (plus one, for consistency reasons regarding the log) over the number of dislikes (plus one, to avoid divisions by zero) i.e., $c = \log\left(\frac{l+1}{d+1}\right)$. There are generally more likes than dislikes and the opposite happens in about only 0.6% of the cases. Interestingly, this index is at best weakly correlated with explicit actions. This may denote an intrinsic property of videos. As for the two extra random walk entropy measures, we find that unlike $\bar{\eta}$, $\bar{\eta}_c$ is positively correlated with $N$ ($\rho = 0.45$), and that $\bar{\eta}_a$ is only very weakly correlated with $N$ ($\rho = 0.14$). The mild positive correlation between $\bar{\eta}_c$ and $N$ is already hinted at by the category coloring of the sample graphs in the lower panel in Fig 4. As already mentioned, mean number of distinct visited nodes per random walk ($\bar{N}_v$) and mean degree ($\langle k \rangle$) are very strongly correlated with $\bar{\eta}$. Finally, we see that age shows close to no correlation with any of the metrics, except for a weak correlation with $v$ ($\rho = 0.27$).

A plausible interpretation for the interplay between random walk diversities (especially $\bar{\eta}$ and $\bar{\eta}_c$), recommendation graph size ($N$) and number of views ($v$), arises from modeling the recommendation engine as a knowledge-discovery process. By viewing a video, the user provides empirical data on the probability of relatedness of the video being watched and all the videos the user has watched before. Of course, there are certainly myriad implementation details on how different signals and pieces of information about the user and the video are taken into

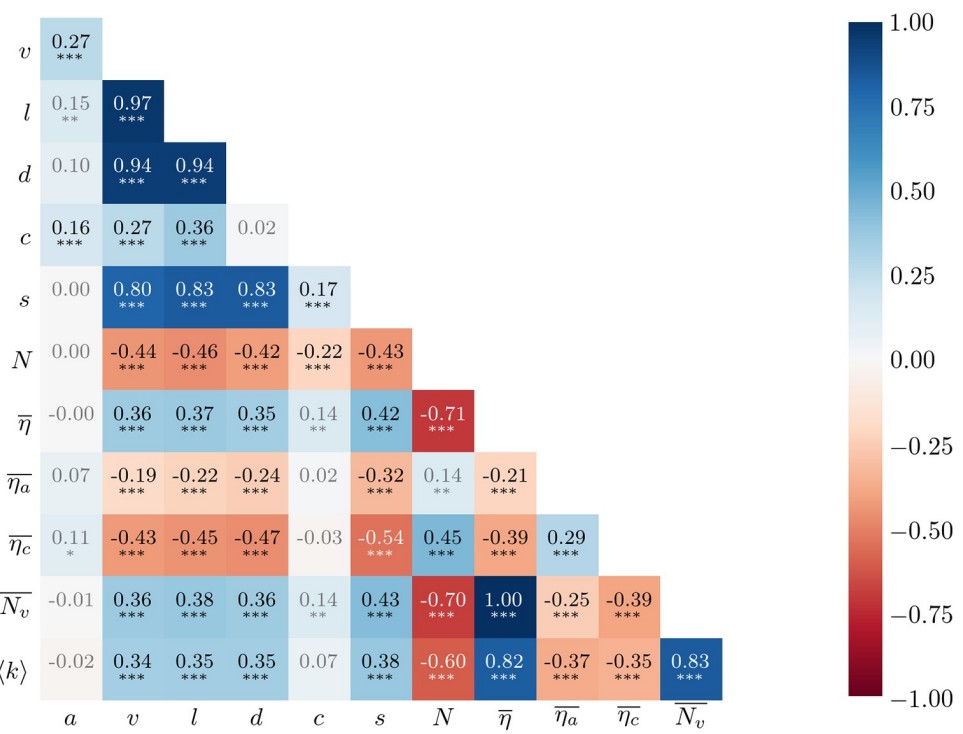

**Fig 6. Pearson correlations between various recommendation graph metrics.** Asterisks indicate significance: *** for $P < 0.0001$, ** for $P < 0.001$, * for $P < 0.01$.

account to tweak the recommendation process. Here we are not interested in reverse-engineering a given recommendation engine, but instead in using empirical data to try to uncover more general dynamics from a user's perspective. This is of particular interest to understand how a generic recommendation engine may mediate the exploration of a given cultural space by human actors. Independently of the details, it appears trivial to assume that users viewing videos also provide a connection between this video and the videos previously seen by them. The observation that the age of a video has almost no correlation with any of the other metrics goes in favor of the interpretation that the exploration by users of the cultural space of YouTube videos may help the platform in focusing and, thereby, confining its recommendations.

This standpoint invites us to take the number of nodes in the recommendation graph as an expression of uncertainty. The user is given more choices, but these choices lead to more constrained paths. A video with more views plausibly induces a better knowledge by the platform of its relatedness to other videoss. Recommendations are thus possibly more focused: smaller in overall number, but more inter-related between themselves, and thus further constraining the user in a general sense, while providing a more diverse navigation path, in terms of distinct video IDs, within this more constrained realm. This interpretation is given further credence by the fact that, even though random walk video diversity $\bar{\eta}$ increases with $N$, random walk category diversity $\bar{\eta}_c$ decreases. In other words, *the user is exposed to a higher diversity of unique videos on a less diverse set of topics.*

## Confinement and transitions

We dig further this notion of topical confinement by focusing on the node level and especially the navigations induced by jumping from a video to another one. More precisely, for each

node that appears in any crawl, we compute the outgoing transition probabilities for immediate recommendations i.e., we examine dyadic directed links from a node to the members of the plateau found for that node. We distinguish three types of features related to topics, on the one hand, and to explicit user actions, on the other hand; all of which are linked to some intrinsic property of a seed video (semantics, popularity, consensus):

- *topical categories*, found in the meta data of the respective videos. We focus on the six top categories in the whole data set (News & politics, Entertainment, Music, People & Blogs, Science & technology, Howto & Style). YouTube provides for many other possible categories which each appear less than a dozen times here, so we gathered them as "[Other]".

- *contentment indices*, defined as before as the log of the ratio of likes over dislikes. Since negative values are rare, we gather them into a single category denoted as "negative". Integer ranges strictly above 4 are also strongly underpopulated (less than a dozen of occurrences each) and are, again, gathered as "[Other]".

- *number of views*, binned as quartiles whose boundaries are $\{143k, 960k, 5.31m\}$ views.

In Fig 7, we show the likelihood of jumping from a video with some property to a recommended video of the plateau with some property as transition matrices. Results are aggregated over all nodes appearing in the various seed-centric crawls.

For one, it appears generally that topical categories are also topological categories, even though we observe large variations across topics: from "Music" which is massively self-reinforcing, to "People & blogs" which rather redistribute users toward other topics, especially "Entertainment".

The effect of "contentment" displays a quite different picture. There are few negatively rated videos and contentment typically ranges between 1 and 3. Yet, there is also a tendency to redistribute users toward videos which are more positively rated so, in a sense, the recommendation landscape does not confine users into controversial areas.

Views follow a rather automorphic tendency where, irrespective of the origin quartile, recommended videos generally exhibit the same order of magnitude as the origin video. This effect is particularly strong for the most viewed videos. As such, the recommendation landscape does not seem to push viewers of less viewed videos towards most viewed videos. Furthermore and similarly, mainstream videos do not appear to forward users towards less viewed videos, which likely induces a reinforcement mechanism in these areas, opposite to the conclusions of [28]. One may suggest that we just observe here the result of an *a posteriori* redistribution mechanisms where videos recommended from the most viewed ones incidentally garner views and end up in the highest quartile as well. This hypothesis is however invalidated by the computation of these transition matrices restrained to newly appearing videos only i.e., videos that were not part of the plateau when collecting recommendation graphs (see below): these matrices do exhibit exactly the same patterns as the ones shown on Fig 7.

In other words and to summarize, following mean-field recommendations, users are incited (1) to navigate within the same topical category, especially so for musical and political/news videos, (2) to remain in sets of videos which have rather comparable numbers of views, especially so for mainstream videos, and (3) to go towards more consensual videos, yet to a lesser extent when videos are moderately consensual.

## Evolution of recommendation graphs and origin of novelty

We previously observed that recommendation sets attached to a seed video slowly evolve with time. New suggestions appear in the plateau over time. We may ask in which direction does

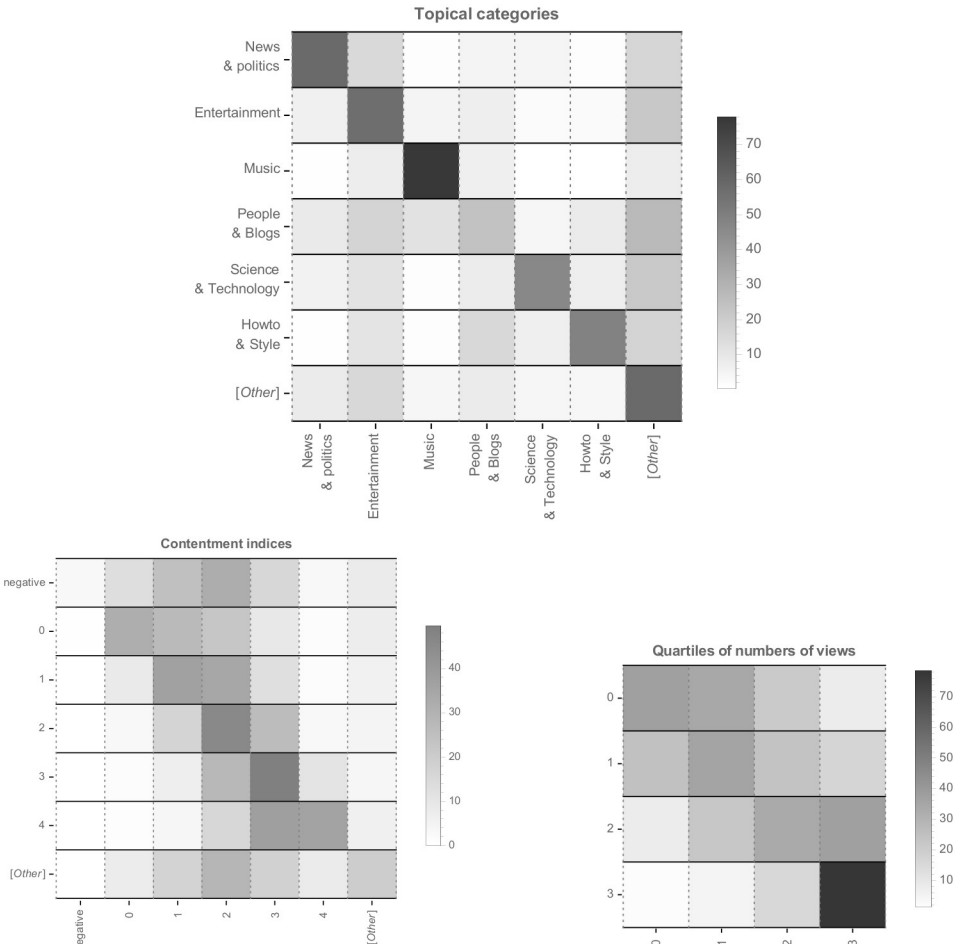

**Fig 7. Recommendation transition matrices for all nodes, with respect to topical categories (*top*), contentment index (*bottom-left*) and quartiles of numbers of views (*bottom-right*).**

the introduction of novelty in recommendation sets alter the picture that we sketched so far and, in particular, where do new suggestions come from and what percentage of them stems from inside vs. outside the known recommendation graph. Put differently, is novelty really novel? To check this, we consider as *novelty* the new plateau suggestions for seed videos appearing at the end of the long crawl i.e., $R = 2000$ requests after the recommendation graph has been collected. We first notice that percentages vary greatly across seed videos, as shown on the left panel in Fig 8: most plateaus nevertheless exhibit at least a third of novel videos, with an average of about 58%. However, many of these novel recommendations can be found not far in the recommendation graph, at depth 2 or 3. In other words, a significant portion of suggestions at $R = 2000$ come from inside the known graph at $R = 0$ (almost four in five): reinforcement is also at work here, in the sense that new suggestions are either direct or indirect neighbors.

Similarly, we could also verify that transitions matrices restricted to novel recommendations are of the same nature as those which were observed in Fig 7: the aggregated matrices look almost indistinguishable from the original matrices (we thus do not shown them here).

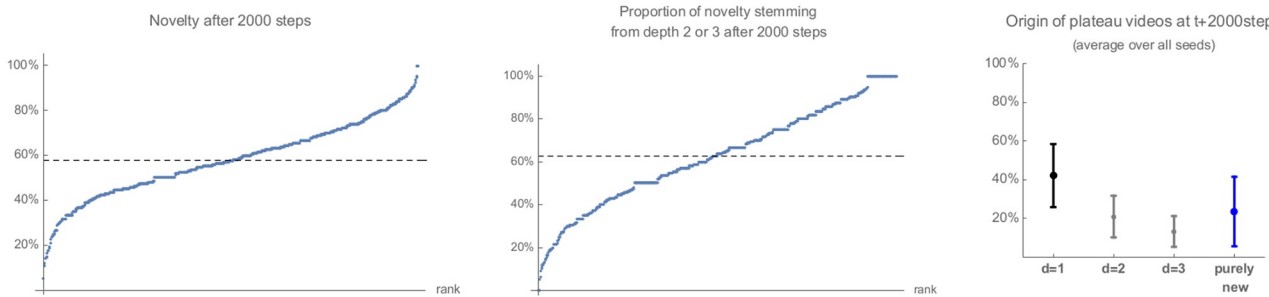

**Fig 8. Provenance of new suggestions for seed videos.** *Left:* Distribution of the percentage of novelty: percentage of plateau recommendations which are new at the end of the long crawl (after $R = 2000$ requests) vs. its beginning. *Middle:* Distribution of the percentage of such novelty which could nonetheless be found deeper in the recommendation graph already at the beginning of the crawl. *Right:* Average, over all seeds, of the provenance of plateau recommendations, with respect to their position in the recommendation graph. Error bars indicate standard deviations.

## Concluding remarks

This work was focused on recommendation graphs extracted from YouTube. Two types of findings were attained: firstly about the temporal dynamics of the mean-field recommendations provided by the platform for a given seed video, and secondly about the configuration of local recommendation graphs centered around seed videos, especially in regard to confinement and diversity. The former does not aim at reverse-engineering: it is purely instrumental to the purpose of the latter. In this respect, we could exhibit a plateau of highly frequently suggested videos and characterize this phenomenon statistically, both in terms of size and duration. This led to an exploration and retrieval protocol that is both computationally feasible and leads to observables—the recommendation graphs—with well-justified and empirically grounded boundaries. Recommendation graphs are, for one, a peculiar sort of networks, with a modal degree distribution.

In turn, the analysis of these graphs according to several metrics, notably measures of confinement, led to a better understanding of recommendation dynamics, including its interaction with users. In a nutshell, be it in topological, topical or temporal terms, the landscape of what we call mean-field YouTube recommendations generally exhibits confinement. However, we could also show that this claim must be nuanced in various directions.

- First, recommendation graphs exhibit a wide range of values of entropies: some graphs are more confined or confining than others. Counter-intuitively, higher entropies (in terms of navigation) are associated with lower diversity (in terms of distinct number of accessible videos). This hints at a dichotomy where some seed videos are at the root of an isotropic navigation (higher entropy) in a more limited space of videos (lower size).

- Second, we could demonstrate that higher entropies are found for seed videos with a higher number of views. We hypothesize that a higher popularity means that more information is being collected and thus plausibly enables the platform to refine the associated recommendation graph. This contributes to hint at a dynamic of increasing confinement driven by user activity: this would have to be further tested with a longitudinal deep crawl, which however remains beyond the scope of this paper.

- Third, we exhibited the existence of confinement in topical terms (topical categories are also topological clusters), temporal terms (seemingly new recommendations are not to be found too far in the recommendation graph), popularity terms (high view videos transition to high

view videos, keeping in mind the correlation between the number of views and topological confinement), but not in contentment terms.

This may come as a surprise when considering the growing state of the art on the question of the diversity of content accessible to users through algorithmic recommendation: so far, this body of work appears to concur on the fact that platforms rather tend to expand the navigation landscape of users—here, the mean field of YouTube recommendations shows up as an exception.

Future work should certainly appraise a variety of other modes of recommendation (such as personalized suggestions), other types of behavior (such as organic navigation, whereby users search for videos by themselves) and a mix thereof (such as browsing on subscription-based channels). On the whole, the analysis of the graphs we extracted nonetheless demonstrate the diversity of navigation anisotropy on YouTube in a variety of dimensions. They also suggest that the most confined graphs i.e., potential bubbles, are organized around videos that garner the highest audience and plausibly viewing time. Admittedly, our work could help devise algorithms that make users aware of their possible confinement, in line with [37] and [38]. While our results further indicate that it is difficult to provide a binary answer to the question of confinement on this platform, they appear to nuance the emerging picture in the literature that implicit recommendation has a neutral or even horizon-expanding role.

## Acknowledgments

We are grateful to Lucie Lamy, Serge Reubi and Ayşe Yuva for their kind contribution to the human-based confinement evaluation step, and to Katharina Tittel for her participation in the Twitter data set perimeter definition. We thank Martin Gerlach for his constructive remarks.

## Author Contributions

**Conceptualization:** Camille Roth, Telmo Menezes.

**Data curation:** Antoine Mazières.

**Formal analysis:** Camille Roth, Telmo Menezes.

**Funding acquisition:** Camille Roth.

**Methodology:** Camille Roth, Antoine Mazières, Telmo Menezes.

**Project administration:** Camille Roth.

**Supervision:** Camille Roth.

**Validation:** Camille Roth, Antoine Mazières, Telmo Menezes.

**Visualization:** Camille Roth, Telmo Menezes.

**Writing – original draft:** Camille Roth, Telmo Menezes.

**Writing – review & editing:** Camille Roth, Antoine Mazières, Telmo Menezes.

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
