## [Decision Letter · Decision Letter 0]

13 Mar 2020

PONE-D-20-02223

Tubes and Bubbles - Topological Confinement of YouTube Recommendations

PLOS ONE

Dear Dr. Roth,

Thank you for submitting your manuscript to PLOS ONE. After careful consideration, we feel that it has merit but does not fully meet PLOS ONE’s publication criteria as it currently stands. Therefore, we invite you to submit a revised version of the manuscript that addresses the points raised during the review process.

We would appreciate receiving your revised manuscript by Apr 27 2020 11:59PM. To enhance the reproducibility of your results, we recommend that if applicable you deposit your laboratory protocols in protocols.io, where a protocol can be assigned its own identifier (DOI) such that it can be cited independently in the future. For instructions see: http://journals.plos.org/plosone/s/submission-guidelines#loc-laboratory-protocols

We look forward to receiving your revised manuscript.

Kind regards,

Tiago P. Peixoto

Academic Editor

PLOS ONE

Journal Requirements:

2. In your Methods section, please include additional information about your dataset and ensure that you have included a statement specifying whether the collection method complied with the terms and conditions for the website.

Additionally, PLOS ONE has been specifically designed for the publication of the results of primary scientific research that address a clearly defined research question, add to the base of knowledge in a scientific discipline, and are presented in an intelligible manner. (http://journals.plos.org/plosone/s/criteria-for-publication). We are concerned that this submission does not meet such criteria. We appreciate that you have developed a protocol to extract reliable video-centric recommendation graphs from YouTube. However, it is our view that the presented problem and methodology to its solution do not seem to be motivated by a clearly outlined research question and that the physical and mathematical insight provided from this analysis remains unclear. As such, we kindly ask you to revise the manuscript to clearly state your objectives, and to define specifically how they will contribute to new insight in the field.

Reviewers' comments:

Reviewer's Responses to Questions

**Comments to the Author**

1. Is the manuscript technically sound, and do the data support the conclusions?

Reviewer #1: Partly

2. Has the statistical analysis been performed appropriately and rigorously? 

Reviewer #1: Yes

3. Have the authors made all data underlying the findings in their manuscript fully available?

Reviewer #1: Yes

4. Is the manuscript presented in an intelligible fashion and written in standard English?

Reviewer #1: Yes

5. Review Comments to the Author

Reviewer #1: In this manuscript the authors approach the problem of how filter bubbles in online social systems can emerge as a result of the underlying recommendation system. For this they investigate the structure of networks obtained from following recommendations across videos on youtube. Specifically, the authors create and validate a new methodology for collecting such data and perform a detailed computational analysis of the obtained networks finding that i) networks contract as the seed videos become more popular, and ii) the dynamics of the random walks exhibit confinement.

This work makes a substantial contribution to recent and ongoing efforts to better understand the dynamics in socio-technical systems such as youtube which have become a major platform for dissemination and consumption of information. I like the innovative methodological approach taken by the authors to look under the hood of these recommendation systems via recommendation networks (and the rigor in validating it), especially given the fact that the underlying algorithms&data are not openly accessible. The statistical anaylsis is solid and the results are presented in an intelligible way. My main point of criticism is on the interpretation of the results in terms of the dynamics (see details below). I think it should be clarified that all conclusions come from correlational analysis and not from following the dynamics of individual videos over time in order to avoid misunderstandings. Pending corresponding revision, I fully recommend publication of the paper.

1. The results of mean entropy and number of pageviews is described as a graph contraction (line 217): "It appears that, as videos receive views, their overall recommendation graphs contract, becoming significantly smaller in number of nodes, while the diversity of the mean random walk increases."

The results in Figure 4 suggest a strong correlation between entropy and number of views. However, the conslusion as written suggests a dynamical process of indivdual videos which would only be justifieds from longitudinal studies.

2. In line 253 the authors refer to "audience effect" - I am not aware of a clear definition and couldnt find one in the manuscript. Perhaps the authors could clarify in order to avoid misunderstanding on what is meant.

3. In line 257 the authors claim that the measure of c is an intrinsic property of the video bc it is not correlated with explicit actions. I think this statement is a bit too strong in the absence of following individual videos over time.

4. In line 278 the authors claim that the dynamics is dominantly driven by the actions of the user. The dataset does not follow the dynamics of a video over time.

6. PLOS authors have the option to publish the peer review history of their article (what does this mean?). If published, this will include your full peer review and any attached files.

Reviewer #1: Yes: Martin Gerlach

---

## [Author Response · Author response to Decision Letter 0]

25 Mar 2020

Dear Editor, Dear Reviewer,

We thank you for your consideration of our manuscript and are very pleased to have carried out the requested revision. We also fully agree with all the remarks of Reviewer 1, whom we wish to thank for his report on our work.

This revision attemps to improve the manuscript by taking into account all the issues that have been raised. In particular, we changed the wording related to the dynamics of recommendation graphs — it is true that confirming the contraction of these graphs over time because of user interaction would require a different protocol, where we would collect and study recommendation graphs in a longitudinal manner. In this respect, we amended our remarks to reflect the fact that we simply observed correlations between views and graph size and density, and that we cannot directly conclude on graph contraction, especially on pages 7, 8, 11 (previously 10), and 13.

We also added a clarification sentence regarding what we mean with “audience effect” on page 10 (previously 9), and toned down several claims identified by Reviewer 1. We further inserted two citations to two additional studies published in 2020 which feature the computation of diversity or confinement through random walks.

More broadly, we also took into account the general remark pertaining to our goals and to the framing of our work within the literature — we added sentences to this effect in the abstract, in the introduction, and in the conclusion. We hope that this further clarifies our positioning within the state of the art on the effects of algorithmic recommendation.

We finally added a precision regarding the respect of YouTube Terms of Service for the data collection operations. We mentioned the availability of our data and scripts in publicly-available repositories.

We emphasized all these changes by putting the text in blue where needed (for the sake of clarity, we however did not emphasize the very minor typo and form corrections).

We hope to have addressed all matters of concern regarding this manuscript in order to make it fit for publication. We nevertheless remain at your disposal to carry out further improvements.

---

## [Decision Letter · Decision Letter 1]

31 Mar 2020

Tubes and Bubbles - Topological Confinement of YouTube Recommendations

PONE-D-20-02223R1

Dear Dr. Roth,

We are pleased to inform you that your manuscript has been judged scientifically suitable for publication and will be formally accepted for publication once it complies with all outstanding technical requirements.

With kind regards,

Tiago P. Peixoto

Academic Editor

PLOS ONE

Additional Editor Comments (optional):

Reviewers' comments:

Reviewer's Responses to Questions

**Comments to the Author**

1. If the authors have adequately addressed your comments raised in a previous round of review and you feel that this manuscript is now acceptable for publication, you may indicate that here to bypass the “Comments to the Author” section, enter your conflict of interest statement in the “Confidential to Editor” section, and submit your "Accept" recommendation.

Reviewer #1: All comments have been addressed

2. Is the manuscript technically sound, and do the data support the conclusions?

Reviewer #1: Yes

3. Has the statistical analysis been performed appropriately and rigorously? 

Reviewer #1: Yes

4. Have the authors made all data underlying the findings in their manuscript fully available?

Reviewer #1: Yes

5. Is the manuscript presented in an intelligible fashion and written in standard English?

Reviewer #1: Yes

6. Review Comments to the Author

Reviewer #1: The revised manuscript addresses all the concerns raised in my previous comments substantially improving the manuscript. I recommend publication of the manuscript in PLOS ONE.

7. PLOS authors have the option to publish the peer review history of their article (what does this mean?). If published, this will include your full peer review and any attached files.

Reviewer #1: Yes: Martin Gerlach

---

## [Editor Report · Acceptance letter]

2 Apr 2020

PONE-D-20-02223R1 

Tubes and Bubbles Topological Confinement of YouTube Recommendations 

Dear Dr. Roth:

I am pleased to inform you that your manuscript has been deemed suitable for publication in PLOS ONE. Congratulations! Your manuscript is now with our production department. 

With kind regards,

on behalf of

Dr. Tiago P. Peixoto 

Academic Editor

PLOS ONE